# Neuroimmunology and Allergic Disease

Sayantani B. Sindher, Vanitha Sampath , Andrew R. Chin, Kari Nadeau and Rebecca Sharon Chinthrajah *

Sean N. Parker Center for Allergy and Asthma Research at Stanford University, Stanford, 240 Pasteur Dr, BMI #1454, Palo Alto, CA 94304, USA
* Correspondence: schinths@stanford.edu

**Abstract:** The prevalence of allergic diseases is rising globally, inducing heavy quality of life and economic burdens. Allergic reactions are mediated by the complex bi-directional cross-talk between immune and nervous systems that we are only beginning to understand. Here, we discuss our current understanding of the molecular mechanisms of how this cross-talk occurs in the skin, gut, and lungs. An improved understanding of the communication between the immune and nervous system may lead to the development of novel therapies for allergic diseases.

**Keywords:** allergy; atopic dermatitis; neuroimmunology



## 1. Introduction

Aberrant cellular and humoral immune responses to innocuous external stimuli in allergic diseases, such as allergic asthma, allergic rhinitis, atopic dermatitis (AD), and food allergies, are caused by well-characterized proinflammatory responses mediated by innate and adaptive immune systems. Proinflammatory allergic sensitizations are initiated by epithelial-derived cytokines (IL-25, IL-33, and thymic stromal lymphopoietin), which activate dendritic cells and drive the differentiation of naïve CD4+ T cells to Th2 cells and the subsequent production of Th2 type cytokines (IL-4, IL-5, IL-9, and IL-13). These type 2 cytokines promote tissue mast cells, basophils, and eosinophil accumulation; IgE class switching by B cells; the production of IgE; and the binding of allergen-specific IgE to FcεRI receptors on mast cells or basophils. This sensitization phase is followed by the effectorphase where subsequent allergen exposure leads to the crosslinking of FcεRI-bound IgE; and the degranulation of mast cells or basophils leading to the release of chemokines, such as histamine and other inflammatory chemical mediators such as cytokines, proteases, interleukins, leukotrienes, and prostaglandins. These mediate physiological and structural changes in target tissues such as the airways and the skin. Allergic reactions can range from mild to severe, systemic or localized, and may include anaphylaxis, sneezing, coughing, wheeze, itch, edema, or vomiting [1].

While the immune system plays a pivotal role in allergic disease, it is now evident that it does not act alone. There is mounting evidence that the peripheral neurons, similarly to cells of the immune system, also directly sense allergens and affect tissue responses via the release of neurotransmitters and neuropeptides (e.g., calcitonin gene-related peptide, substance P and acetylcholine) and also via cross-talk with immune cells. Neurons actively communicate with and regulate the function of immune cells, such as mast cells, dendritic cells, eosinophils, and type 2 innate lymphoid cells through the expression of surface membrane receptors that recognize and bind chemical mediators released by immune cells-neurotransmitters and neuropeptides by neuronal cells and cytokines and other inflammatory mediators by immune cells [2]. In recent years, the presence of anatomically distinct neuro-immune cell units (NICUs), in which both immune and neuronal cells colocalize and interact to coordinate responses to environmental stimuli, has been described in peripheral tissues such as the skin, gut, and brain, which have been identified [3]. In this review, we discuss the neuroimmune interactions that mediate allergic inflammation.

## 2. Neuroimmune Cross-Talk in the Skin

In allergy diseases, the skin mounts an immune response to innocuous allergens, and these inflammatory responses are measured as wheal sizes and form the basis of diagnosis during a skin prick test. Furthermore, when an allergen comes in contact with the skin, the abundant nociceptors in the skin transduce the signal to electrical activity, producing itching through cross-talk with mast cells, basophils, eosinophils, dendritic cells, ILC2s, and Th2 cells (Figure 1).

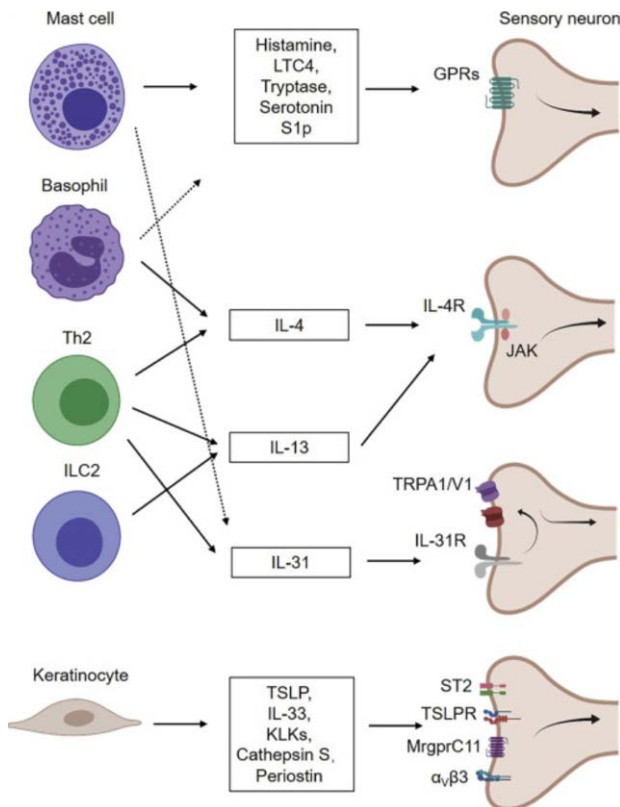

**Figure 1.** Immune and epithelial cells release cytokines and proteases that can activate itching sensations. Mast cells are a major source of histamine and leukotrienes such as tryptase, and sphingosine-1 phosphate (S1p), which can act directly on sensory neurons via GPCRs. IL-4 and IL-13 directly stimulate sensory neurons and promote itching via IL-4Rα-JAK1 signaling on sensory neurons. Both TRPA1 and TRPV1 are involved in the downstream itch pathways mediated by IL-31. The epithelial cytokines thymic stromal lymphopoietin (TSLP) and IL-33, which can also directly stimulate sensory neurons via their receptors TSLPR and ST2, respectively, mediate itch. Dotted arrows indicate limited evidence. (Copyright permission to reproduced this Figure has been obtained from [4]).

Mast cells are closely associated with nociceptors where they can form specialized neuroimmune clusters that regulate immune responses [5–9]. Mouse studies have demonstrated that allergens, such as house dust mites, trigger nociceptors expressing *Tac1*, the precursor of neuropeptide substance P (SP), to release SP, which activates the Mas-related G protein-coupled receptor B2 (Mrgprb2) in mast cells [6]. The activation of Mrgprb2 leads to the secretion of histamine as well as cytokines and chemokines such as TNFα, GM-CSF, IL-8, CCL2, CCL3, and CCL4 to trigger IgE-independent allergic inflammation and mast-cell degranulation (Figure 2) [6,10]. These findings combined other studies demonstrate that although SP is canonically associated with signaling through the neurokinin-1 receptor (NK-1R), SP-induced mast-cell degranulation predominantly occurs via Mrgprb2/MRGPRX2 (the human ortholog of Mrgprb2) [6,8–12]. In addition to the role of MRGPRX2 in contact allergic reactions, MRGPRX2 is also responsible for anaphylactoid and pseudo-allergic reactions to drugs [9,13–16]. Therefore, there is mounting interest in the development of

drugs that block the MRGPRX2 activation of mast cells for the treatment of itch and other allergic processes [17]. The neurotoxin capsaicin has been found to increase skin mast-cell activity in mice [18] and potentially decreasing the total number of mast cells in the bladder of rats [19]. The interaction between nociceptors and mast cells is not limited to allergic responses and further discussions of this cross-talk in non-allergic pathologies can be found in a recent review [20].

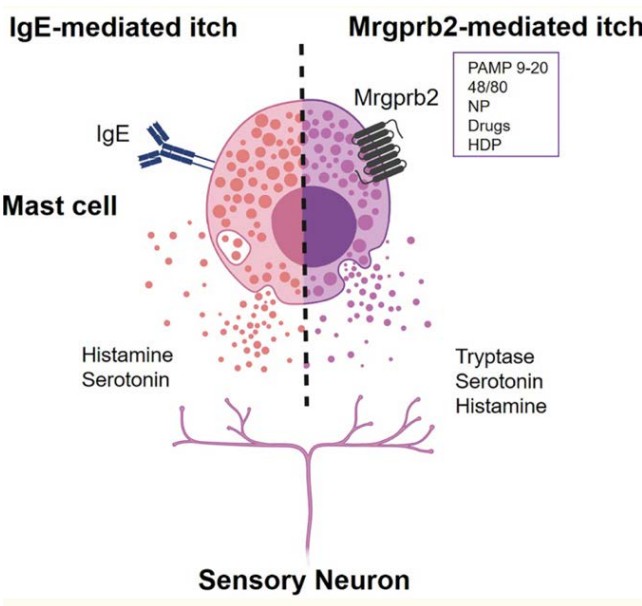

**Figure 2.** The IgE-mediated versus Mrgprb2-mediated itch axis. The classical activation of mast cells by IgE results in the release histamine and serotonin. Mrgprb2 (murine)/MRGPRX2 (human) can be activated by various cationic substances, such as pro-adrenomedullin peptide 9–20 (PAMP 9–20), compound 48/80, drugs, neuropeptides (NPs), and host defense peptides (HDPs). The Mrgbprb2-mediated activation of mast cells elicits distinct mechanisms of itch from classical IgE stimulation, in which tryptase is a major mediator while others such as histamine and serotonin are also included. (Copyright permission to reproduced this Figure has been obtained from [5]).

Another neuropeptide implicated in AD is the calcitonin gene reactive protein (CGRP). Late-phase skin reactions in AD subjects were associated with the infiltration of inflammatory cells expressing neuropeptide CGRP, suggesting that these vasoactive factors might play a role in the erythema and edema characteristics of allergic inflammation [21].

Histamine, produced by mast cells and basophils, is the best characterized pruritogen and plays an important role in allergic inflammation. It acts on four distinct G protein-coupled receptors (H1R-H4R). Of these, H1R and H4R have important roles in the progression and modulation of histamine-mediated allergic diseases. They are expressed on histaminergic nerves in various tissues and, upon binding, histamine activates transient receptor potential (TRP) ion channels on sensory neurons, resulting in membrane depolarization, a subsequent action potential, and itching sensations. An in vitro study of murine sensory dorsal root ganglia neurons found that TRPV1 inhibition led to a reduction in H1R- and H4R-induced itch, whereas TRPA1 inhibition reduced H4R- but not H1R-induced itching [4]. TRPV4 is another member of the TRP family implicated in itching, and the pharmacological blockade of TRPV4 attenuates calcium responses within DRGs exposed to histamine. Antihistamines that block the activation of H1R have been shown to be effective therapeutics for the treatment of pruritus associated with urticaria, allergic rhinitis, and allergic conjunctivitis. However, their efficacy in AD is limited. A phase 2a, randomized, double-blind, placebo-controlled, multicenter, parallel-group study of a H4R-antagonist (JNJ-39758979) in Japanese adults with moderate AD showed promising although inconclusive results in alleviating pruritus in AD patients but was terminated early due to adverse

effects [22]. Recently, synergistic effects were observed with the combined treatment with H1R and H4R antagonists with a reduction in Th2 inflammatory responses in the nasal mucosa of rats with allergic rhinitis [23]. Future studies are needed to determine the effectiveness of the combined H1R and H4R antagonists in AD.

IL-4 and IL-31 are produced by various immune cells such as Th2 cells, dendritic cells, and eosinophils and are implicated in itches and inflammation (Figure 3). IL-31 signals via a heterodimeric receptor composed of IL-31 receptor A (IL-31RA) and oncostatin M receptor β, both of which are expressed in dorsal root ganglion neurons. Elevated levels of IL-31 or its receptor have been reported in the tissue or serum of patients with AD and are correlated with SCORAD AD severity scores [24]. Nemolizumab, a new monoclonal antibody that targets IL-31RA, has shown efficacy in AD in two long-term phase 3 trials [25]. In a study by Oetjen et al., the authors demonstrate that the activation of IL-4Rα directly stimulates human sensory neurons and that the intrinsic activation of the sensory neuron of this signaling pathway (IL-4Rα and JAK1) mediates chronic itching. Furthermore, JAK inhibitor treatments ameliorate symptoms in patients with chronic itching [26].

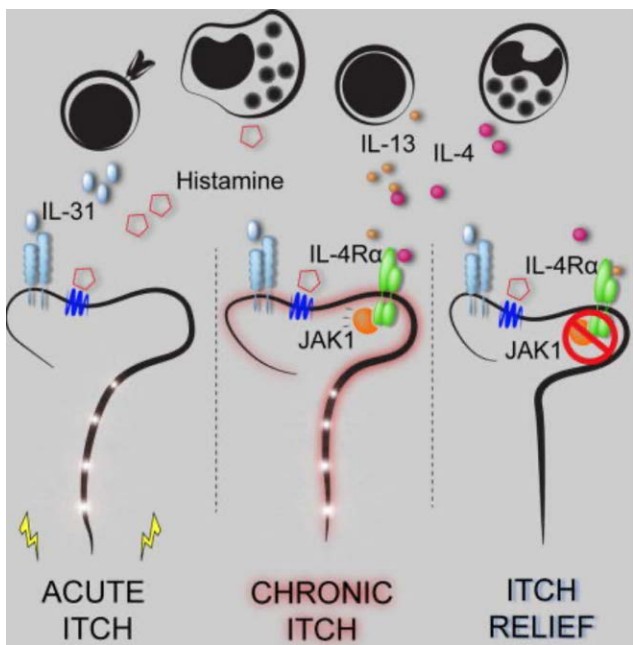

**Figure 3.** Acute and Chronic itch: IL-31 mediates acute itching via IL-31RA. The activation of IL-4Rα mediates chronic itch via JAK1. The inhibition of JAK1 ameliorates the symptoms (Copyright permission to reproduced this Figure has been obtained from [26]).

### 3. Neuroimmune Cross-Talk in the Gut

Mucosal mast cells can be found in close proximity to calcitonin gene-related peptide immunoreactive (CGRP-IP) neurons in food allergic mice [27]. The stimulation of CGRP-IP neurons induces microtubule reorganization in mucosal mast cells, promoting a flattened spread morphology that enhanced the degranulation of mucosal mast cells in response to other stimuli [28,29]. In addition, enteric neurons also express high-affinity IgE receptors (FcεRIs) for which its stimulation can lead to the activation of mucosal mast cells to potentially exacerbate food allergy pathology [30]. However, further studies are needed to better characterize these interactions in allergic diseases.

### 4. Neuroimmune Cross-Talk in the Lungs

In the airways of patients with asthma and allergic rhinitis, several neurotransmitters and neuropeptides increased. Brain-derived neurotrophic factor (BDNF) and its high-affinity BDNF receptor tyrosine kinase receptor B (TrkB) are higher in asthmatic airway smooth muscle cells than in non-asthmatics. It contributes to airway remodeling by enhanc-

ing extracellular matrix production and deposition, especially of collagen-1 and collagen-3 [31]. BDNF may, therefore, present a novel approach attenuating remodeling in diseases such as asthma. Nerve growth factor (NGF) is another neurotropin that has been associated with asthma pathogenesis. Increased levels of NGF are found in asthma patients, and it is associated with disease severity. NGF is associated with eosinophils survival by inhibiting its apoptosis and the expression of NGF increased in eosinophils [32]. In a rat model of chronic asthma, NGF exacerbates allergic lung inflammation and airway remodeling [33]. In a mouse model of chronic experimental asthma, anti-NGF or anti-TrkA antibody treatments reduced collagen depositions in the airways [34]. In an ovalbumin-induced rat asthma model, a nebulized formulation of anti-NGF inhibited airway remodeling [35]. Twenty-four hours after nasal provocation, BDNF and NGF were upregulated in the nasal mucosa and increased in the peripheral blood of patients with allergic rhinitis [36].

Both parasympathetic and sympathetic neurons are involved in modulating allergic immunity and inflammation in the respiratory tract. Acetylcholine (Ach) is the main neurotransmitter released by the parasympathetic nervous system. It is released from the vagus nerve and activates muscarinic ACh receptors (mAChRs 1 through 5), leading to bronchoconstriction, increased mucus secretion, inflammation, and airway remodelling [37]. Of the five identified muscarinic receptors, only M1, M2, and M3 receptors have been shown to play major roles in airway physiology. Acetylcholine is also released from non-neuronal cells such as airway epithelial cells and other immune cells. The effects on inflammation and remodeling are regulated by both neuronal and non-neuronal acetylcholine [38]. M3 receptor antagonists are currently used as bronchodilators for the treatment of asthma.

Noradrenalin is released by sympathetic nerves and is involved in the bronchial muscle relaxation of smooth muscle cells via the $\beta_2 AR$ receptor. It is expressed widely on many types of immune cells such as ILC2s. In a mouse model of asthma, $\beta_2 AR$ signaling inhibited the activation of ILC2 and decreased it in key inflammatory cytokines [39]. In a mouse model of allergic asthma, norepinephrine was found to stimulate IgE production on binding β2-adrenegic receptors and activating B cells [40].

## 5. Conclusions

A positive feedback loop between neuronal and immune cells exists; however, our understanding of these interactions and the signals that mediate their responses to the ever-changing physiological and pathological conditions is still very limited. Understanding the reciprocal interactions between the immune and neuronal cells presents a new frontier and a challenge in understanding allergic diseases.

**Funding:** No funding was received for the production of this manuscript.

**Institutional Review Board Statement:** Not applicable.

**Informed Consent Statement:** Not applicable.

**Data Availability Statement:** Not applicable.

**Conflicts of Interest:** Sindher reports grants from NIH, Regeneron, DBV Technologies, AIMMUNE, Novartis, CoFAR, grants and personal fees from FARE, other from Astra Zeneca and DBV; Nadeau reports grants from National Institute of Allergy and Infectious Diseases (NIAID), National Heart, Lung, and Blood Institute (NHLBI), National Institute of Environmental Health Sciences (NIEHS), and Food Allergy Research & Education (FARE); stock options from IgGenix, Seed Health, ClostraBio, and ImmuneID; is Director of the World Allergy Organization Center of Excellence for Stanford, Advisor at Cour Pharma, Consultant for Excellergy, Red tree ventures, Eli Lilly, and Phylaxis, Co-founder of Before Brands, Alladapt, Latitude, and IgGenix; and National Scientific Committee member at Immune Tolerance Network (ITN), and National Institutes of Health (NIH) clinical research centers, outside the submitted work; patents include, "Mixed allergen composition and methods for using the same," "Granulocyte-based methods for detecting and monitoring immune system disorders," and "Methods and Assays for Detecting and Quantifying Pure Subpopulations of White Blood Cells in Immune System Disorders." Chinthrajah receives grant support from the Consortium for Food Allergy Research (CoFAR), National Institute of Allergy and Infectious Disease (NIAID), Food Allergy

Research & Education (FARE), Aimmune, DBV Technologies, Astellas, Novartis, Regeneron, and Astra Zeneca, and is an advisory board member for Alladapt Immunotherapeutics, Novartis, Sanofi, Allergenis, Intrommune Therapeutics, and Genentech. All other authors indicate no COI.

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
