# Peer review of "Neuroimmunology and Allergic Disease"

_allergies, doi:10.3390/allergies2030008_

Round 1
Reviewer 1 Report
I am very much impressed with the paper. The quality of scientific and educational content is very high. In my opinion, the presented for my review paper meets all requirements requested for publication in the Journal "Allergies".

Author Response
We thank the reviewer for taking the time to review this manuscript.
Reviewer 2 Report
The topic, herein presented, is very interesting although it seems that it is still at its early infancy of understanding.
Comments:
The manuscript is well-written, but some punctual missing use of prepositions can be found. Please verify.
Line 24 - IgE are by definition antibodies, which makes "IgE antibodies" a language redundance. Please delete antibodies. Check for similar situations along the manuscript.
Line 25 - substitute "activation" by "elicitation" or "effector phase".
Fig 1- with "itch" do the authors mean pruritus? Probably pruritus would be a better choice of word. Please consider change here and along the manuscript when applicably
Fig. 1 - confirm that permission to publication was asked to the journal. Format citation according to journal guidelines.
Lines 65-66 - the sentence is not understandable. Please correct.
Fig. 2 - same comment as before regarding permission and citation format. Please verify.
Line 112 - present the acronym AD on the first time that is used.
Fig 3 -same comment as fig 1 and 2. Please verify.
Line 128 - presentation of AD should be made some lines before. Please correct.
Section 4 - do the authors have any idea if patients under specific neurological medication may be at increased risk of developing food or other allergic reactions? If possible, comment on this.
Author Response
We thank the reviewer for taking the time to review this manuscript. Our point by point responses are in blue:
The manuscript is well-written, but some punctual missing use of prepositions can be found. Please verify.
We thank the reviewer for pointing this out and made several grammar corrections throughout the manuscript
Line 24 - IgE are by definition antibodies, which makes "IgE antibodies" a language redundance. Please delete antibodies. Check for similar situations along the manuscript.
We have removed “antibodies” from instances of “IgE antibodies” and screened the manuscript for other locations of language redundancy.
Line 25 - substitute "activation" by "elicitation" or "effector phase".
The requested change has been made (lines 25-26).
Fig 1- with "itch" do the authors mean pruritus? Probably pruritus would be a better choice of word. Please consider change here and along the manuscript when applicably
We thank the reviewer for pointing this out and have made the requested change to the Figure legend.
Fig. 1 - confirm that permission to publication was asked to the journal. Format citation according to journal guidelines.
Permission to publish was received from the journals for the use of Figures 1-3. We have updated the citation formatting.
Lines 65-66 - the sentence is not understandable. Please correct.
We thank the authors for pointing this out, we have fixed the typos in this sentence it now reads: “Mast cells are closely associated with nociceptors where they can form specialized neuroimmune clusters that regulate immune responses”.
Fig. 2 - same comment as before regarding permission and citation format. Please verify.
Permission to publish was received from the journals for the use of Figures 1-3. We have updated the citation formatting.
Line 112 - present the acronym AD on the first time that is used.
We have now defined the acronym AD at first use and made consistent use of the abbreviation throughout the manuscript.
Fig 3 -same comment as fig 1 and 2. Please verify.
Permission to publish was received from the journals for the use of Figures 1-3. We have updated the citation formatting.
Line 128 - presentation of AD should be made some lines before. Please correct.
We have now defined the acronym AD at first use and made consistent use of the abbreviation throughout the manuscript.
Section 4 - do the authors have any idea if patients under specific neurological medication may be at increased risk of developing food or other allergic reactions? If possible, comment on this.
At this time we are not aware of any specific neurological medications that may increase the risk of developing food allergy or cause allergic reactions, however we believe that this is an interesting area of research that merits further investigation.
Reviewer 3 Report
The topic is interesting and current. However, the manuscript is rather short and synthetic. A better division among allergic disease should be preferable. The section concerning treatment/modulation of mediators: e.g., capsaicin. References should be as updated as possible.
Author Response
We thank the reviewer for their insights. At this time the allergic neuroimmunology field is still in development and since neuroimmune crosstalk mechanisms are likely shared between allergic diseases, we organized the manuscript by organ system as opposed to disease. We have added a short discussion of capsaicin to the manuscript and updated the references.